# Qualitative cross-country comparison of whether, when and how people diagnosed with lung cancer talk about cigarette smoking in narrative interviews

Senada Hajdarevic,[1] Birgit H Rasmussen,[2,3] Trine L Overgaard Hasle,[4] Sue Ziebland[5]

[1]Department of Nursing, Umeå University, Umeå, Sweden
[2]The Institute for Palliative Care, Lund University and Region Skåne, Lund University, Skåne University Hospital, Lund, Sweden
[3]Department of Clinical Sciences Lund, Surgery, Lund, Sweden
[4]Research Unit of General Practice, Research Centre for Cancer Diagnosis in Primary Care, Aarhus University, Aarhus, Denmark
[5]Health Experiences Research Group, Nuffield Department of Primary Care Health Sciences, University of Oxford, Oxford, UK

**Correspondence to**
Professor Sue Ziebland;
sue.ziebland@phc.ox.ac.uk

## ABSTRACT

**Objectives** To compare and examine whether, when and how patients with lung cancer in three countries, with different survival rates, talk about cigarette smoking and its relationship with help-seeking.

**Design** A qualitative cross-country comparison with analysis of narrative interviews.

**Setting** Participants in Sweden, Denmark and England were interviewed during 2015–2016. Interviews, using a narrative approach, were conducted in participants' home by trained and experienced qualitative researchers.

**Participants** Seventy-two men and women diagnosed with lung cancer were interviewed within 6 months of their diagnosis.

**Results** The English participants, regardless of their own smoking status, typically raised the topic of smoking early in their interviews. Smoking was mentioned in relation to symptom appraisal and interactions with others, including health professionals. Participants in all three countries interpreted their symptoms in relation to their smoking status, but in Sweden (unlike England) there was no suggestion that this deterred them from seeking care. English participants, but not Swedish or Danish, recounted reluctance to consult healthcare professionals with their symptoms while they were still smoking, some gave up shortly before consulting. Some English patients described defensive strategies to challenge stigma or pre-empt other people's assumptions about their culpability for the disease. A quarter of the Danish and 40% of the Swedish participants did not raise the topic of smoking at any point in their interview.

**Conclusion** The causal relationship between smoking and lung cancer is well known in all three countries, yet this comparative analysis suggests that the links between a sense of responsibility, stigma and reluctance to consult are not inevitable. These findings help illuminate why English patients with lung cancer tend to be diagnosed at a later stage than their Swedish counterparts.

## INTRODUCTION

Health problems which are known to be associated with certain elective 'health behaviours' can lead to stigma and victim blaming.[1] A prominent example is lung cancer (LC).

### Strengths and limitations of this study

► The study design was informed by social science theories about stigma, shame and blame in the context of lung cancer.
► Discussion of smoking was not prompted by the interviewers' questions, adding strength to our comparisons of whether, when and how participants raised the topic.
► This analysis is based on an initially unanticipated point of comparison between the three countries, underlining the benefit of using an unstructured narrative approach to interviewing and attending to emergent themes during qualitative analysis.
► The results are based on patients' accounts and we did not observe patients' interactions with healthcare professionals or other people.
► While relatively large for a qualitative study we only interviewed participants in three countries, all of which have publicly funded health systems.

Patients in England and Scotland have reported that, on hearing the diagnosis, other people assume that they are (or have been) heavy smokers.[2–5] Images of blackened lungs and television adverts including the personal testimonies of patients with LC voicing their regrets for smoking add to the impression of personal culpability.[3] A highly cited qualitative study published in the *BMJ* drew attention to the 'stigma, shame and blame' that many people with LC experience due to the disease's association with smoking.[3] This matters for several reasons, including because studies have shown that patients who fear being blamed are deterred from seeking help when they have symptoms.[3 4] Yet, as we will show, there is no inevitability about the link between perceptions of responsibility, experience of stigma and a reluctance to consult. We draw on findings from our comparative qualitative study of experiences of LC in Sweden, Denmark and England.

LC is the largest cancer-related cause of death worldwide.[6] Denmark, UK and Sweden have similar population smoking rates (~18%, 20% and 20%, respectively).[7] International Cancer Benchmarking data have demonstrated that patients with LC in Denmark and England are diagnosed at a more advanced stage than those in Sweden.[8] Sweden also continues to show a higher 1 year survival for LC than Denmark and England.[9]

Possible explanations for why people in England and Denmark consult with later stage disease have been considered, including differences in how people interpret bodily sensations, access healthcare, describe their problem to their doctors and how the doctors respond. It has also been demonstrated that English and Danish smokers with LC symptoms consult their general practitioners (GP) later than non-smokers with the same symptoms.[9–11]

Sociocultural aspects and characteristics of the healthcare system influence how and when people seek care.[12] Past experiences of contact with the healthcare services contribute to expectations and subsequent decisions to seek help.[13–18] Awareness of LC as a smoking-related disease is high in all three countries. Smoking in restaurants, bars and workplaces has been banned in Sweden since 2005, Denmark since 2007 and England since 2007, with the legislation reinforcing the reduced social acceptance of smoking.

This paper draws on in-depth interviews collected during a cross-country qualitative comparative study of the lead-up to a cancer diagnosis in three northern European countries.

The wider study sought to illuminate international variations in cancer stage at diagnosis by comparing people's accounts of their decisions and experiences of initially seeking care.[19] The current paper explores whether, when and how these interviewees with LC talked about cigarette smoking during their open-ended, narrative interviews. The topic of smoking was always raised by the participant, not the research team, enabling us to compare when and how they framed their talk about lung cancer and smoking.

## METHOD
### Participants and recruitment
We conducted a cross-country comparative qualitative analysis of interviews collected within 6 months of a diagnosis of lung cancer (LC) in Denmark (n=22), England (n=20) and Sweden (n=30). Interviews were conducted during 2015–2016 as part of a wider qualitative comparative study which also included interviews with patients with bowel cancer as well as LC.[19] Purposive sampling[20] through hospital clinics (in Sweden) and also primary care support organisations via snowball sampling and social media (in England and Denmark) sought comparable variation across gender, age, urban and rural locations and pathway to diagnosis.

### Data collection
In the wider comparative study participants gave informed consent and decided the place for their interview, usually their own home. The experienced qualitative researchers had backgrounds in nursing (Sweden), sociology (England) and anthropology (Denmark). A semistructured interview guide based on social science theories, and cancer research literature,[3 15 21–25] was developed to enhance cross-country comparisons.[26] The multidisciplinary fieldwork team maintained close monthly contact by telephone conference during data collection. The whole team also met on several occasions for data analysis workshops which were held in all three countries. Further consideration of the challenges and solutions involved in cross-country qualitative work, drawing on examples from this project, have been reported.[26] All interviews began with an open-ended question: '*Could you start by telling me, in your own words and in as much detail as you want, about what has happened since you first started to suspect there might be a problem with your health?*' followed by flexibly used prompts from an interview guide. We deliberately avoided specific questions about whether the person smoked, or had ever smoked, cigarettes, thus smoking status was always volunteered by the patient or marked as 'not known'. Interviews lasted between 45 and 90 min and were audio recorded for transcription and analysis.

### Analysis
A specialist computer software (NVivo V.10) was used to code, sort and retrieve data. The interview transcripts were read several times before coding. The cross-country differences in talk about smoking arose as an unanticipated theme at one of our first analysis workshops when the principal investigator (a coauthor on the 2004 'Stigma, Shame and Blame' paper[3]) became aware that the centrality of smoking in British narratives about LC appeared to be less evident in the Swedish and Danish interviews. As a first step to test whether this impression was valid, we noted *whether* and *how early* in the interview the topic of smoking was first raised by the participant. Second, drawing on modified grounded theory and the technique of constant comparison we used thematic analysis[27] to consider *how* the participants talked about smoking during their interviews. People are described as 'current,' 'ex-smokers' or 'recent ex-smokers' on the basis of their narrative accounts. The researchers neither asked about smoking nor made assumptions about the smoking status—hence we describe smoking status as 'not known' if it was not raised by the participants.

Three authors (SH, SZ and BHR) used a data-driven coding framework developed through discussion at face-to-face workshop meetings and via phone conferences. We discussed and refined our interpretations of observed similarities and differences between the countries. For this paper, direct quotes have been translated into English (and checked by BHR, the trilingual member of our team).

## Patient involvement

Public and patient involvement (PPI) was conducted in accordance with good practice in each country. Patients with LC in England were involved in the whole process from the preparation of funding application through the development of the study and discussion of preliminary results. PPI members in all three countries commented on the participant information sheets and the interview guide and advised on recruitment strategies.

## RESULTS

The researchers' impression that there were differences between countries in the frequency, positioning and content of talk about smoking was confirmed when we compared the point in the interview at which participants raised the topic of smoking. Our thematic analysis of the talk about smoking draws on theories of responsibility,[1] shame and blame[3] and is presented in relation to symptom appraisal, help-seeking and communication with others.

### Whether and how early smoking was raised by the participant

All but two of the English participants raised the topic of smoking, unprompted, in their interviews; this usually occurred within the opening sections or first 15% of the interview transcript. More than one-quarter of the Danish participants and 40% of the Swedish participants did not raise the topic of smoking at all in their interviews; those who did were less likely than their English counterparts to raise it in the early part of their narrative (table 1).

**Table 1** Participant characteristics and cross-country comparison of *whether* and *how early* in the interview the topic of smoking was raised by the participant

| | Denmark | England | Sweden |
|---|---|---|---|
| Total number of interviewed participants | 22 | 20 | 30 |
| Women (%) | 8 (36) | 10 (50) | 15 (50) |
| Age range (year) | | | |
| 31–50 | 0 | 2 | 2 |
| 51–70 | 15 | 12 | 21 |
| 71–90 | 7 | 6 | 7 |
| Participants who raised the topic of smoking within the first 15% of the interview (%) | 9 (41) | 11 (55) | 9 (30) |
| Participants who raised the topic of smoking at any point in their interview (%) | 16 (73) | 18 (90) | 18 (60) |
| Smoking NOT raised by participant during interview (%) | 6 (27) | 2 (10) | 12 (40) |

## How participants talked about smoking

Participants raised the topic of smoking when talking about how they had interpreted their symptoms and when describing interactions with others, including how their healthcare professionals responded.

### Smoking as a frame to interpret symptoms

Our first observation is that participants often raised the topic of smoking as if answering an (unasked) question, for example: *'But I mean, if I hadn't of broken me back they wouldn't have found (the cancer). And [er…pause]. The question is, 'Did you stop smoking?' well, Yeah'* (Ex-smoker, ENG ID18).

When describing decisions to consult, those who told us that they had never smoked or said they were longer term ex-smokers raised their (non)smoking status as part of their explanation for not expecting this diagnosis.

I thought of course that it was pneumonia or something, so I never thought…as we all know, I have never smoked, I have never worked in smoky surroundings. I could never imagine that it could be something like that. (Never smoker, SWE ID147)

Long-term ex-smokers often commented they had not been expecting LC because they had stopped smoking many years ago. '*I mean, it never dawned on me that I was—lung cancer, didn't really figure. I have smoked when I was younger. That was 34 years ago*' (Ex-smoker, ENG ID12).

People who were still smokers, or who told us they had recently stopped, said they had interpreted and normalised their symptoms—a long-term cough was seen as a 'smoker's cough' and a degree of breathlessness as normal for a smoker.

I had this persistent cough which I recognised as a smoker's cough, like [coughs] and it was constant. (Ex-smoker, ENG ID05)

You know that you smoke so it's normal with breathlessness when one was walking and exerted oneself a bit too much. So that was something you were expecting since you were smoking. So, therefore, you did not react in some specific way to it. (Current smoker, SWE ID105)

And as long as I was smoking I was also coughing. But it was…it did not get worse. And it also disappeared when I quitted smoking, so I really did not have any indication. (Ex-smoker, SWE ID150)

Smokers also justified a reluctance to consult on the grounds that the sensations and symptoms they experienced were not sufficiently serious, that they felt too well, or the signs were not those they associated with LC.

I always thought lung cancer was due to smoking. [um] And I knew people who had lung cancer and they used to say to me, you know, [um] coughing up a lot of blood, things like that. I went, 'Really.' And I didn't have none of that. You see. So I never put it down to cancer. (Current smoker, ENG ID09)

I know that I'm smoking and I've been smoking for many years, but I've always had good health and I've never coughed like many do in the morning for many years… or… I have never done that. So, I did not really think it could be cancer, because I had it great, and in principle I still have. (Current smoker, DK ID14)

### Smoking as a frame when interacting with others

Accounts from some of the English smokers suggested that healthcare professionals who appeared to 'blame' smoking for the symptoms had delayed taking action.

… I said to the GP that my Dad and my Uncle had lung cancer and I thought they should have been a bit quicker to check it out. … Basically it [the symptoms] was worse at night time and everything. First of all I think she [the GP] blamed it on my smoking. (Current smoker, ENG ID03)

We also heard several accounts, in all three countries, of healthcare professionals who had worked hard to avoid any suggestion of blame or responsibility in relation to smoking.

The doctor said—'You have to let it go—the cancer—you can get it whether you smoke or not, you can get it from passive smoking. Not to say that it is just smoke, you can get it just from the air and anything else'—So I should let it go and not hit myself in the head with it. (Current smoker, DEN ID17)

I said, 'Have I caused this cancer through smoking?' So she started laughing. She [the doctor], said 'I'll explain, this type of cancer that you've got it's in women, more in women than in men.' She said, 'And it's not smoke related because people that don't smoke get this cancer more than people that do smoke so, no.' (Current smoker, ENG ID09)

In the following example, a Swedish ex-smoker who voiced her concern that smoking might have been responsible was contradicted, kindly, though not entirely convincingly, by the doctor.

'Now we should not mix in smoking in this,' the doctor said. And they were saying that in the order that I should not have such a guilty conscience, I can imagine. But she said that there were many non-smokers who get lung cancer,… but obviously they [doctors] think it [smoking] might have contributed, absolutely. (Recent ex-smoker, SWE ID123)

While at least some of the smokers (and ex-smokers) in all countries alluded to feeling partly responsible for their LC, this was far more dominant in the English participants' accounts. English participants also described feeling the need to pre-empt and manage people's assumptions that the LC must be smoking related. One man, who had not smoked for over four decades, explained that he had labelled his disease a 'non-smoking lung cancer' to distance himself from people's assumptions.

I can be classed as a non-smoker, 45 years no smoking. So we found out that, you know, lung cancer does have a stigma to it in society. I went down, I just had been chatting to neighbours two doors down about what had happened and she said, 'Oh, that will stop you smoking, won't it.' So we always describe it as 'non-smoking lung cancer.' (Ex-smoker, ENG ID02)

Another described her pre-emptive tactic of explaining that she had stopped smoking a long time ago.

Yeah. I think there is a little element […], that you feel you know, you were stupid to smoke in the first place and you shouldn't have done. But then, look how many people smoke so [laughs]. Yeah, so, yeah, there is that little hint of, 'Well it's your own fault' sort of thing. So I'm very quick to say I haven't for 34 years. (Ex-smoker, ENG ID12)

Interestingly, while none of the Swedish or Danish participants said that they had stopped smoking shortly before consulting the GP, several of the English smokers said they gave up smoking before they consulted. The following participant became convinced that something was wrong when the breathlessness did not abate after stopping smoking:

I gave up smoking. I had the, you know, something isn't right. I gave up smoking and then I got a chest infection. …It was when I went back to the GP after the chest infection because I still couldn't breathe. That underlying breathlessness was still there. It was then that I just, something tells me that something's not right because I'm still feeling breathless and I packed in smoking. (Recent ex-smoker, ENG ID01)

## DISCUSSION

Our study shows that, while there is high awareness of LC as a smoking-related disease in all three countries, the 'smoking' frame is a more immediate and dominant part of the LC narrative in England. In the English narratives, accounts of LC are infused with references to smoking. Only in England did current smokers describe a reluctance to consult because of their smoking status or stopping smoking before they consulted the doctor. Our study draws attention to cross-country differences in how cigarette smoking is framed in relation to discourses of responsibility[1] for LC. The relationship between responsibility, blame and stigma reported by Chapple et al[3] in 2004 appears to be mediated by the health system and culture, even between seemingly comparable northern European countries. Stigma, shame and blame matter, because they affect how the individual copes with their diagnosis, and because they affect when and how people present their symptoms. There is no evidence to suggest that English participants were more aware of the relation between smoking and LC than their Scandinavian counterparts, but only in the English accounts was this awareness offered as a reason for not consulting with symptoms.

Earlier studies from England have shown that smoking contributes to delayed care seeking[2 4 15 18 28] and that smokers seek care late due to feelings of shame and personal responsibility.[15 18 29] Recent studies from England and Denmark also show that smokers are less likely than non-smokers to seek care for symptoms such as cough or hoarseness.[30 31] Our comparative study suggests that the primacy given to smoking in the accounts mirrors cultural differences and norms about the connection between perceived blame and help-seeking. The narratives suggest that in Sweden the smoking status of the individual is less relevant to the decision to seek care; indeed the 'Goldilocks Zone', within which patients feel it is appropriate to access to healthcare services,[32] may be broader in Sweden than in England.

Healthcare professionals who avoid victim blaming and treat the patient with kindness may influence people's attitudes to seeking care.[17 33] As Corner and Brindle argue,[34] social processes are the means of culture and social organisation, shaping how people interpret and present symptoms to healthcare professionals, and affecting timely cancer diagnosis.

We also found more accounts of managing social interactions defensively, among participants from England, some of whom described acting defensively when others made assumptions about the cause of their LC. This was seldom mentioned by the Danish and Swedish participants. There are comparable findings from a Swedish study of patients diagnosed with another smoking-related disease, the chronic obstructive pulmonary disease.[35]

### Strengths and weakness of the study

This qualitative study with a cross-country comparison contributes to knowledge about the cultural context of health, care seeking and smoking among people recently diagnosed with LC. A strength of this study is that the study design was informed by social science theories about stigma, shame and blame in the context of LC. The interviews produced rich data from three northern European countries with different LC survival rates; the team approach enabled careful comparative analysis. Discussion of smoking was not prompted by the interviewers' questions, adding strength to our comparisons of whether, when and how participants raised the topic. This analysis is based on an initially unanticipated point of comparison between the three countries, which would not have been evident had a more structured approach to the interviews been used, or had smoking status been among the standard questions. This underlines the benefit of using an unstructured narrative approach to interviewing and attending to emergent themes during qualitative analysis.

Our study has limitations. While relatively large for a qualitative study we only interviewed participants in three countries, all of which have publicly funded health systems. The data were gathered as part of a larger study, the main results of which have been published in this journal.[19] The results are based on patients' accounts and we did not observe patients' interactions with healthcare professionals or other people. However, as Greenhalgh[36] argues, people's stories about illness are a critical window to meaning systems and values and are nested with the narratives of society and culture.

### CONCLUSION

By demonstrating differences between patients' narratives of LC in England, Sweden and Denmark the study challenges the notion that there is any inevitability about the link between responsibility, stigma and reluctance to seek care. There is nothing inevitable about whether and how this influences the patients' decision to consult, or how an individual's history of smoking cigarettes might affect the conduct of the consultation itself. Rather, the link between perceived responsibility and help-seeking seems to reflect particular relations between the public and publicly funded healthcare. The causal relationship between smoking and LC is undeniable, yet the consequences of this awareness for patients seeking care are not uniform.

The study sheds light on findings from international comparisons of cancer survival by offering a new, potentially modifiable, explanation for why patients with LC in England may be diagnosed at a later stage.

**Acknowledgements** We are extremely grateful to the people who took part in this research, and to the study advisory panel, including patient and public representatives, who helped design the study and provided comments on an earlier draft of this manuscript. We also acknowledge the support of the National Institute for Health Research, through the Clinical Research Network, who helped recruit patients into the English arm of the study. We also thank all those who helped to recruit participants. In Denmark, we thank the Lung Cancer Patient organisation for assisting us in recruiting patients. In England, we thank the NHS Hospital Trusts that assisted with this study and Patients Active in Research: Thames Valley, along with the following charities who posted links or circulated our details on social media: Beating Bowel Cancer, Bowel Cancer UK, Roy Castle Lung Cancer Foundation, British Lung Foundation, and healthtalk.org. In Sweden, we thank the nurses and physicians who helped with recruitment. Interviews in England were conducted by John MacArtney. Professor Ziebland is an NIHR Senior Investigator. We dedicate this paper to our colleague and coapplicant Alison Chapple PhD (1944–2018) whose work inspired this analysis. Alison was a lifelong non-smoker who was diagnosed with lung cancer in 2017, shortly after her retirement.

**Contributors** SZ developed the idea for the manuscript. SH wrote the first draft, with edits made by SZ, BHR and TLOH. SH, SZ, BHR and TLOH contributed to the analysis meetings and provided comments on drafts. SZ prepared the final draft and responded to the reviewer's comments.

**Funding** This paper presents independent research funded by organisations from three European countries as follows: In the UK, the study was supported by the National Awareness and Early Diagnosis Initiative (NAEDI), http://www.naedi.org.uk. Project Award C7663/A17663. The contributing partners include: Cancer Research UK; Department of Health, England; Economic and Social Research Council; Health and Social Care Research and Development Division, Pubic Health Agency, Northern Ireland; National Institute for Social Care and Health Research, Wales; and the Scottish Government. This funding also covered the costs associated with the comparative analysis meetings in Denmark and Sweden and funded translation of the Danish and Swedish material for publications. In Denmark, the study was supported by the Research Centre for Cancer Diagnosis in Primary Care funded by The Danish Cancer Society and the Novo Nordic Foundation. In Sweden, the study was supported by the Vårdal Foundation; the Strategic Research Program in Care Sciences (SFO-V), Umeå University; the Cancer Research Foundation in Northern Sweden; and from government funding of clinical research within the National Health Service, Sweden.

**Disclaimer** The views expressed in this paper are those of the authors, and not necessarily those of the NAEDI, Danish and Swedish funding partners.

**Competing interests** None declared.

**Patient consent** Not required.

**Ethics approval** This study was approved by the ethical boards in each country separately in accordance with their requirements: in England by Research Ethics Service (reference 14/NS/1035); in Sweden by Regional Ethics Board, Lund, Sweden (registration number 2014/819), and in Denmark, the Biomedical Research Ethics Committee System Act does not apply to this project, as the project does not implicate the use of human biological materials. Standard ethical protocol according to the American Anthropological Association was followed.

**Provenance and peer review** Not commissioned; externally peer reviewed.

**Data sharing statement** All researchers in each country had access to all the data gathered in their own country. Interview transcripts were not translated and were not shared between countries, but Swedish and Danish interview extracts were translated by the bilingual researchers for the analytic discussions. The team lead in each country (SZ, RSA and BHR) takes responsibility for the integrity of the data and the accuracy of the data analysis. In Denmark, the data are available for secondary analysis in conjunction with members of the research group named Comparative Cancer Experiences. In England, the participants gave informed consent for data to be copyrighted to the University of Oxford for secondary analysis, broadcasting, publication and teaching. In Sweden, the data are available for secondary analysis in conjunction with members of the research group named Comparative Cancer Experiences.

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
