## [Reviewer comments · BMJ Open]

ARTICLE DETAILS

TITLE (PROVISIONAL)	A qualitative cross-country comparison of whether, when and how people diagnosed with lung cancer talk about cigarette smoking in narrative interviews
AUTHORS	Hajdarevic, Senada; Rasmussen, Birgit; Overgaard Hasle, Trine; Ziebland, Sue

VERSION 1 – REVIEW

REVIEWER	Julie Walabyeki Hull York Medical ~School University of Hull Cottingham Road HU6 7RX
REVIEW RETURNED	20-May-2018

GENERAL COMMENTS	This is an interesting qualitative paper comparing talk on cigarette-smoking and help-seeking among recently diagnosed lung cancer patients in Denmark, Sweden and England. The findings provide explanations to why the English lung cancer patients present later to the doctor than their Swedish counterparts. The strengths and limitations of the paper are also provided. While this paper is novel and original, there are several issues to address in order to strengthen the paper. 'What is already known on this subject': Please revise the first point-add when those cigarette smoking rates were obtained and revise the next sentence too. Introduction: This section needs revision because the research question/aim of the study is not clearly presented nor is the study rationale. Consider revising this section. Methods: A better description of this section including recruitment, data collection and analysis is required. I recommend using a checklist to report the study findings using for example, COREQ (Tong et al 2007) or SRQR (O'Brien et al 2014). Consider adding a flow diagram to illustrate the study recruitment, inclusion and exclusion criteria. The paper does not mention how the smoking status of the interviewees was obtained although it clearly states that specific questions on smoking were avoided. How was 'recent ex-smokers' defined? Results: Consider adding the interviewee smoking status details to Table1; ensuring that the excerpts are labelled uniformly and consider adding gender. Page 10: Please revisit the first excerpt, it is not clear. Discussion: Responsibility has been mentioned in this section however there is no literature/references indicated anywhere in the paper. Please revise the last paragraph which is not clear. Furthermore, please expound on how this links to private/publically funded healthcare. Are there any areas for future research?
---

	References: There is a typographical error in reference number 7.
REVIEWER	Sara Golden VA Portland Health Care System; USA
REVIEW RETURNED	16-Jul-2018
GENERAL COMMENTS	1) include more information about the informed consent process and methods. For instance, if the interviewers avoided questions about smoking, how was the study posed in the consent? Did the participants know they were being asked for their views about smoking or was there a form of deception and debriefing used? Also, how long were the interviews? Can you provide a table of characteristics of participants (race, gender, education, etc.)? It might also be helpful to define how participants are labeled in the manuscript (i.e., ENG= England, LC=?, L=?) 2) I'd like to see more in the results about the reluctance of UK patients to consult their GP. As written there is only one quote that does not make it clear that most of the participants reported delaying help seeking due to their feelings of blame. 3) The authors conclude on pg. 14 line 18-19 that it is the relations between public and publically funded healthcare, but I'm curious if there is research showing that this would be different in a private healthcare system? This does not need to be added to the manuscript since it may be outside the scope, but it might be interesting to see if these feelings equated to a greater or lesser likelihood of quitting smoking for those who were current smokers.

VERSION 1 – AUTHOR RESPONSE

Introduction: This section needs revision because the research question/aim of the study is not clearly presented nor is the study rationale. Consider revising this section.

We have removed the initial summaries of 'what is known and what this adds' and incorporated these points into the main body of the paper. We have clarified the research question and its relationship to the wider study (already published in BMJ open, reference 19) in the introduction

Methods: A better description of this section including recruitment, data collection and analysis is required. I recommend using a checklist to report the study findings using for example, COREQ (Tong et al 2007) or SRQR (O'Brien et al 2014).

We have added detail to the methods and completed the SRQR checklist as requested

Consider adding a flow diagram to illustrate the study recruitment, inclusion and exclusion criteria
The paper does not mention how the smoking status of the interviewees was obtained although it clearly states that specific questions on smoking were avoided. How was 'recent ex-smokers' defined?

We have clarified these points, adding further explanation and annotation to clarify that this analysis is based on a narrative interview study in which the interviewers avoided raising the topic of smoking - all reports of smoking status were self reports

Results: Consider adding the interviewee smoking status details to Table1; ensuring that the excerpts are labelled uniformly and consider adding gender.

As above - we have clarified this, changed the title and added a line to the table. gender (% women) was already included

Page 10: Please revisit the first excerpt, it is not clear.
we believe this is clearer in the context of the further explanations we have included

Discussion: Responsibility has been mentioned in this section however there is no literature/references indicated anywhere in the paper. Please revise the last paragraph which is not clear. Furthermore, please expound on how this links to private/publically funded healthcare. Are there any areas for future research?

We have included the references to 'responsibility' which were in the introduction. We have also revised the discussion and conclusion to incorporate the sections removed from 'what this study adds' and believe it is now considerably clearer

References: There is a typographical error in reference number 7.
corrected

Reviewer: 2

1) include more information about the informed consent process and methods. For instance, if the interviewers avoided questions about smoking, how was the study posed in the consent? Did the participants know they were being asked for their views about smoking or was there a form of deception and debriefing used?

thank you - we have now explained this

Also, how long were the interviews?
We have added this to methods (45-90 mins)

Can you provide a table of characteristics of participants (race, gender, education, etc.)?
We have included the data that we think is relevant to the study in table 1

It might also be helpful to define how participants are labeled in the manuscript (i.e., ENG= England, LC=?, L=?)

thank you for pointing out this inconsistency, which we have corrected

2) I'd like to see more in the results about the reluctance of UK patients to consult their GP. As written there is only one quote that does not make it clear that most of the participants reported delaying help seeking due to their feelings of blame.

The reluctance to consult was not the focus for this paper but has been discussed in earlier work (including our own) which is referenced We have expanded and clarified the introduction and discussion sections and repeated the references to the relevant background papers.

3) The authors conclude on pg. 14 line 18-19 that it is the relations between public and publically funded healthcare, but I'm curious if there is research showing that this would be different in a private healthcare system?

This does not need to be added to the manuscript since it may be outside the scope, but it might be interesting to see if these feelings equated to a greater or lesser likelihood of quitting smoking for those who were current smokers.

Thank you. We have slightly revised the discussion and conclusion - to our knowledge there is no other literature that has considered this question in relation to different health systems and none comparing public and privately funded systems

VERSION 2 – REVIEW

REVIEWER	Julie Walabyeki Hull York Medical School, University of Hull, England
REVIEW RETURNED	07-Sep-2018

GENERAL COMMENTS	The revised version is clear and concise. I am not sure I would repeat the study using the above methods section since there is no inclusion and exclusion criteria presented.
--

REVIEWER	Sara Golden VA Portland Health Care System, USA
REVIEW RETURNED	24-Aug-2018

GENERAL COMMENTS	Thank you for the opportunity to review the revision of this manuscript. I believe it is much stronger because of the changes and I appreciate the authors' attention to addressing the previous comments. I recommend this article for publication.
--